# Biostimulant Effects of Waste Derived Biobased Products in the Cultivation of Ornamental and Food Plants

Enzo Montoneri [1,*], Andrea Baglieri [1] and Giancarlo Fascella [2]

1   Dipartimento di Agricoltura, Alimentazione e Ambiente, Università di Catania, 95123 Catania, Italy; abaglie@unict.it
2   CREA Research Centre for Plant Protection and Certification, 90011 Firenze, Italy; giancarlo.fascella@crea.gov.it
*   Correspondence: enzo.montoneri@gmail.com

**Abstract:** Soluble bio-based substances (SBS) may be isolated from the anaerobic digestate of the organic humid fraction of urban waste; from the whole vegetable compost made from gardening residues and from the compost obtained after aerobic digestion of a mixture of urban waste digestate, gardening residues and sewage sludge. These SBS can be used as sustainable and efficient plant biostimulants in alternatives to the commercial products based on fossil sources such as the Leonardite. The present review summarizes the main findings obtained from recent studies accomplished with the SBS applied on several ornamental (Euphorbia; Lantana; Murraya; Hibiscus) and vegetable species (tomato; red pepper; spinach; maize; bean; wheat; tobacco; oilseed rape) with the aim to evaluate their effect on plant growth; fruit and ornamental quality. The main results from these studies show that the non-commercial SBS are more efficient than commercial fossil-based products; at equal applied doses; in enhancing plant growth; leaf chlorophylls; photosynthetic activity; fruit ripening and yield and aesthetic effect; improving flower and fruit quality and optimizing water use efficiency. Depending upon the plant species, increases of the plant performance indicators ranging from zero to 1750% are reported for the plants cultivated in the presence of SBS, relatively to the control plants cultivated in absence of SBS added to the cultivation substrate. The review suggests that biowaste recycling is a sustainable and environmentally friendly source of plant biostimulants, as an alternative to existing fossil sourced agrochemicals.

**Keywords:** bio-based substances; biostimulants; antifungal agents; municipal biowastes; ornamental plants; food plants

## 1. Introduction

Cultivating plants is a human activity involving several sectors. Agriculture deals with cultivation of crops for human consumption as well as animal production. Horticulture strictly involves the cultivation of plants for food consumption, as well as plants not for human consumption. Horticulture differs from floriculture. The former involves different types of garden crops, while the latter involves flowering and foliage plants. Ornamental horticulture is the cultivation of decorative plants of all kinds, including not only plants with attractive flowers, but also plants with decorative leaves, stems, bark, or fruit. Basically, floriculture and ornamental horticulture have decorative and aesthetic purposes. Aside from categories' definitions and differences in the cultivated species, all these categories' activities share similar problems.

Common farming practice is to boost plant production with a fertilizer dose higher than that adsorbed by soil and plant. Thus, noxious fertilizers' components accumulate in soil, reach the food chain, leach through soil into ground water, and ultimately affect human and animal health. Mineral and organic fertilizers are used. The global fertilizer market is 156 billion USD/year. [1]. Major ones are urea and mineral phosphates (80% of the EU fertilizers' market value), with 0.11–0.46 €/kg production cost. They are based on

energy-intensive production processes or manufactured from non-renewable feedstock imported from third countries [2]. Organic fertilizers belong to a niche market (0.15% of the total fertilizer market) [3]. The world consumption of mineral fertilizers containing N, P and K is ca. 200 Mt/year [4]. EU consumption of mineral fertilizers is 16 Mt/year [5]. From 70 to 250 kg/ha nitrates leaching may occur depending on fertilizer dose, soil, and plant type [6]. Based on average 51 kg/ha applied surplus and total 175 Mha cultivated area, 9 kt/year nitrate leach through soil and water. To improve the balance between fertilizers dose and crop requirement, the max EU ruled dose is 150–350 kg/ha. Major organic fertilizers are composts of biowastes from urban, animal, or agriculture sources, manure, peat and leonardite hydrolysates. Composts are commonly applied to soil at 10–30 t/ha.year [7]. High doses that obtain the desired effects are due to compost insolubility causing slow nutrients' uptake by plants. This causes leaching of excess major and trace metal components through soil and water. Similarly, manure is applied at 70 t/ha dose. In addition to leaching, manure causes greenhouse-gases emission due to fermentation in soil. For example, typical aerial $NH_3$ concentration in a pig farm is 5–35 ppm against a 25 ppm threshold level [8]. A higher $NH_3$ level harms both animal and human health. Emission of 420,000 t/year $NH_3$ is estimated from a total 1400 Mt/year EU manure production [9]. Peat and leonardite hydrolysates contain soluble organic and mineral matter. EU consumption is 240 kt/year. These hydrolysates are obtained from fossil source. Based on average 40% C content, their use causes 355 kt/year $CO_2$ emission from fossil C and depletion of fossil sources. Except for municipal biowastes (MBW), a common problem of all fertilizers is that their sources are found in restricted sites, not available worldwide. This poses the problem of product supply and cost. The problem is highly relevant in Europe, which imports most of its mineral consumption from third countries.

One other important restraint on plant productivity is pests and diseases. These are highly relevant for food plants. Food production loss due to plant diseases is estimated to be 10–50%/year [10]. Plant protection relies on pesticides use, which increases food cost and may cause hormonal disruption in human. A common problem of all fertilizers is the need to use them together with pesticides. Together with lowering cost, there is much concern for decreasing the exploitation and depletion of natural resources to produce fertilizers.

In the last ten years, relevant research has been focused on bio-stimulants. These belong to a new functional product category (FPC6) contemplated in the New European Fertiliser Regulation [11,12]. Bio-stimulants are supposed to stimulate the plant metabolism, regardless of their nutrient content, and so improve plant growth, even under abiotic stress, and resistance to diseases [13]. Applied at much lower doses than common mineral and organo-mineral fertilisers, bio-stimulants are expected to induce plant resistance to pathogens and provide high crop productivity at the same time. In this fashion, they reduce/minimize the negative environmental impact of the excessive application of commercial fertilisers and pesticides.

Work by the authors in the past fifteen years proves that soluble bioorganic substances (SBS) obtained from urban and agriculture biowastes have both biostimulant and antifungal properties [14]. No other known products have both properties. The research hypothesis was that, by virtue of the solubility properties and organo-mineral composition, the SBS could increase plant growth and crop production compared to current commercial mineral and organo-mineral agrochemicals. The present paper reports the critical review of work performed by the authors with SBS for the cultivation of food and non-food plants. The review proves the research hypothesis. It demonstrates the SBS biostimulant properties for all tested plants and discusses the economic and environmental benefits for agriculture and horticulture.

## 2. SBS Composition and Properties

The SBS are obtained by hydrolysis at 60–90 °C and pH 13 of several different mixes of urban food, green and sewage sludge wastes fermented under anaerobic and aerobic conditions [14]. Under these conditions, the SBS were obtained together with the secondary

insoluble (IR) product. The fermented wastes yielding the SBS described in the present review were sampled from different streams of the Italian ACEA Pinerolese MBW treatment plant. The SBS contain organic and mineral matter. The organic matter is a mix of molecules with molecular weight from 5 to over 750 kDa. These molecules are constituted by several different organic moieties made by aliphatic and aromatic C substituted by acid and basic functional groups of different strengths. Mineral elements of groups 1 to 4 are bonded to or complexed by the organic moieties. These chemical features are inherited from the pristine biowastes. The molecules contained in SBS are water soluble memories of the native recalcitrant lignocellulosic polysaccharides, proteins, fats, and lignin proximates still present in the biowastes after anaerobic and aerobic fermentation. It is no wonder that, due to their origin, richness of mineral elements, organic functional groups and acquired water solubility, the SBS molecules exhibit a wide range of properties as plant biostimulants, plant resistance inducers, bio-photosensitizers, oxidation catalysts, polymers for manufacturing mulch films, composite pellets, composite plastic articles, and high performance surfactants. Tables 1–4 report the compositional details of the SBS, IR and the pristine fermented biowastes. Table 5 list the plants cultivated with the SBS and summarizes the main SBS effects on the cultivated plants. All data in Tables 1–5 are extrapolated from the references cited in Table 5. The data reported in Tables 1–4 were obtained through a specifically designed analytical protocol [14]. This included calculation of moisture, ash and volatile solids (VS) contents from the sample weight losses determined after heating to 105 and 650 °C, inorganic elements analysis by AAS and/or ICP, microanalyses for C, H, N determination performed with a C. Erba (Rodano, Milan, Italy) NA-2100 elemental analyser. The C types and functional groups reported in Table 4 were determined by solid-state 13C NMR spectroscopy. Solid-state 13C NMR spectra were acquired at 67.9 MHz on a JEOL GSE 270 spectrometer equipped with a Doty probe. The cross-polarization magic angle spinning (CPMAS) technique was employed, and for each spectrum, about 104 free induction decays were accumulated. The pulse repetition rate was set at 0.5 s, the contact time at 1 ms, the sweep width was 35 KHz, and MAS was performed at 5 kHz. Signals assignment as a function of the resonance range were: 0–53 ppm aliphatic C, 53–63 ppm O-Me or N-alkyl C, 63–95 ppm O-alkyl C, 95–110 ppm di-O-alkyl C, 110–140 ppm aromatic C, 140–160 ppm phenol or phenyl ether C, 160–185 ppm carboxyl C, and 185–215 ppm ketone C.

**Table 1.** Waste ingredients in pristine biowastes (PFB).

| PFB | Ingredients |
|-----|-------------|
| D | Digestate from anaerobic fermentation of unsorted food wastes |
| CV | Compost of private gardening and public park trimming residues (V) |
| CVD | Compost of D and V mix in 2/1 weight respective ratio |
| CVDF | Compost of D, V and sewage sludge (F) mix in 5.5/3.5/1 respective ratio |
| ETP | Exhausted tomato plants at the end of the crop harvesting season |

**Table 2.** Mineral elements and ash content (*w/w*%) in the pristine biowastes (PFB), in the soluble (SBS) product and insoluble (IR) hydrolysates obtained.

|  | Si | Fe | Al | Mg | Ca | K | Na | Ash |
|---|---|---|---|---|---|---|---|---|
| CVDF PFB | 6.27 ± 0.04 | 1.02 ± 0.01 | 1.06 ± 0.02 | 0.83 ± 0.01 | 3.23 ± 0.05 | 1.32 ± 0.03 | 0.07 ± 0.01 | 59.4 |
| CVDF SBS | 0.92 ± 0.03 | 0.53 ± 0.02 | 0.44 ± 0.02 | 0.49 ± 0.01 | 2.59 ± 0.03 | 5.49 ± 0.04 | 0.15 ± 0.01 | 27.3 |
| CVDF IR | 7.68 ± 0.06 | 1.23 ± 0.03 | 1.05 ± 0.01 | 1.15 ± 0.02 | 3.20 ± 0.03 | 1.32 ± 0.02 | 0.04 ± 0.01 | 77.6 |
| D PFB | 3.46 ± 0.05 | 0.77 ± 0.03 | 0.40 ± 0.02 | 0.88 ± 0.02 | 7.16 ± 0.08 | 0.53 ± 0.03 | 0.22 ± 0.02 | 34.5 |
| D SBS | 0.36 ± 0.03 | 0.16 ± 0.00 | 0.78 ± 0.04 | 0.18 ± 0.01 | 1.32 ± 0.05 | 9.15 ± 0.06 | 0.39 ± 0.01 | 15.4 |
| D IR | 4.73 ± 0.03 | 0.48 ± 0.01 | 0.47 ± 0.06 | 1.07 ± 0.02 | 9.54 ± 0.05 | 3.44 ± 0.05 | 0.16 ± 0.01 | 49.0 |
| CVD PFB | 10.70 ± 0.03 | 1.07 ± 0.02 | 0.71 ± 0.03 | 1.12 ± 0.01 | 4.27 ± 0.14 | 1.09 ± 0.03 | 0.08 ± 0.01 | 56.1 |
| CVD SBS | 2.49 ± 0.04 | 0.88 ± 0.02 | 0.60 ± 0.06 | 0.93 ± 0.02 | 4.70 ± 0.08 | 3.76 ± 0.07 | 0.17 ± 0.01 | 28.3 |
| CVD IR | 12.60 ± 0.05 | 0.95 ± 0.01 | 0.75 ± 0.03 | 1.13 ± 0.02 | 4.96 ± 0.05 | 2.13 ± 0.06 | 0.07 ± 0.01 | 56.8 |
| CV PFB | 12.14 ± 0.07 | 1.03 ± 0.02 | 0.59 ± 0.01 | 1.67 ± 0.25 | 4.86 ± 0.61 | 1.18 ± 0.07 | 0.06 ± 0.01 | 57.1 |
| CV SBS | 2.55 ± 0.01 | 0.77 ± 0.04 | 0.49 ± 0.04 | 1.13 ± 0.06 | 6.07 ± 0.38 | 3.59 ± 0.21 | 0.16 ± 0.01 | 27.9 |
| CV IR | 15.04 ± 0.33 | 1.10 ± 0.05 | 0.67 ± 0.01 | 1.45 ± 0.01 | 4.19 ± 0.09 | 1.49 ± 0.02 | 0.06 ± 0.01 | 71.3 |
| ETP PFB | 0.98 ± 0.03 | 0.30 ± 0.02 | 0.27 ± 0.02 | 0.42 ± 0.02 | 4.65 ± 0.03 | 3.30 ± 0.02 | 0.22 ± 0.01 | 20.2 |
| ETP SBS | 0.22 ± 0.03 | 0.33 ± 0.02 | 0.34 ± 0.03 | 0.80 ± 0.04 | 2.10 ± 0.02 | 9.15 ± 0.06 | 0.24 ± 0.01 | 23.3 |
| ETP IR | 0.85 ± 0.03 | 0.25 ± 0.01 | 0.17 ± 0.01 | 0.27 ± 0.01 | 4.41 ± 0.02 | 4.49 ± 0.06 | 0.15 ± 0.01 | 36.9 |

**Table 3.** Total C, N and P content (*w/w*%) in pristine biowastes (PFB), and in soluble (SBS) and insoluble (IR) hydrolysates obtained.

|  | C | N | C/N | $P_2O_5$ |
|---|---|---|---|---|
| CVDF PFB | 24.36 ± 0.16 | 2.25 ± 0.11 | 10.83 | 1.30 ± 0.22 |
| CVDF SBS | 35.47 ± 0.09 | 4.34 ± 0.17 | 8.17 | 1.44 ± 0.03 |
| CVDF IR | 11.72 ± 0.22 | 1.02 ± 0.05 | 11.49 | 0.53 ± 0.05 |
| D PFB | 29.99 ± 0.20 | 3.81 ± 0.12 | 7.87 | 3.27 ± 0.15 |
| D SBS | 45.07 ± 0.12 | 7.87 ± 0.12 | 5.73 | 1.14 ± 0.10 |
| D IR | 27.68 ± 0.08 | 1.80 ± 0.05 | 15.38 | 2.75 ± 0.03 |
| CVD PFB | 27.07 ± 0.78 | 2.45 ± 0.07 | 11.05 | 0.75 ± 0.05 |
| CVD SBS | 37.51 ± 0.04 | 4.89 ± 0.03 | 7.67 | 0.84 ± 0.04 |
| CVD IR | 22.11 ± 0.24 | 1.64 ± 0.01 | 13.48 | 1.14 ± 0.18 |
| CV PFB | 22.43 ± 0.42 | 1.91 ± 0.03 | 11.74 | 0.39 ± 0.02 |
| CV SBS | 38.25 ± 0.09 | 4.01 ± 0.03 | 9.54 | 0.53 ± 0.05 |
| CV IR | 18.44 ± 0.67 | 1.15 ± 0.09 | 16.03 | 0.37 ± 0.02 |
| ETP PFB | 36.44 ± 0.24 | 3.51 ± 0.18 | 10.38 | |
| ETP SBS | 47.30 ± 0.09 | 6.52 ± 0.13 | 7.25 | |
| ETP IR | 28.83 ± 0.08 | 2.52 ± 0.10 | 11.44 | |

**Table 4.** Carbon types and functional groups content (*w/w*% of total C) [a].

| | Cal | OMe + NR | OR | OCO | Ph | PhOY | COX | CO |
|---|---|---|---|---|---|---|---|---|
| CVDF PFB | 31.81 | 8.59 | 27.67 | 6.18 | 10.72 | 5.90 | 8.17 | 1.96 |
| CVDF SBS | 31.17 | 7.88 | 19.13 | 6.73 | 16.58 | 7.69 | 10.49 | 0.34 |
| CVDF IR | 28.90 | 8.32 | 27.14 | 7.46 | 13.23 | 7.01 | 6.79 | 1.16 |
| D PFB | 33.60 | 9.10 | 26.61 | 5.99 | 8.94 | 4.27 | 10.53 | 0.97 |
| D SBS | 43.38 | 9.86 | 14.01 | 3.37 | 9.60 | 3.23 | 15.89 | 0.66 |
| D IR | 50.80 | 5.52 | 18.95 | 4.00 | 8.54 | 3.28 | 7.23 | 1.68 |
| CVD PFB | 37.25 | 9.75 | 28.14 | 4.35 | 8.03 | 5.20 | 6.67 | 0.62 |
| CVD SBS | 40.90 | 7.34 | 14.18 | 3.85 | 12.27 | 5.97 | 12.92 | 2.56 |
| CVD IR | 31.73 | 9.39 | 29.32 | 6.39 | 9.78 | 6.21 | 5.87 | 1.31 |
| CV PFB | 32.86 | 8.33 | 23.85 | 6.34 | 12.30 | 6.73 | 8.21 | 1.37 |
| CV SBS | 36.90 | 7.24 | 13.22 | 4.18 | 13.39 | 6.84 | 13.53 | 4.69 |
| CV IR | 31.70 | 8.43 | 24.58 | 6.14 | 11.49 | 7.23 | 7.74 | 2.68 |
| ETP PFB | 14.34 | 7.22 | 49.60 | 11.62 | 6.89 | 3.44 | 6.28 | 0.61 |
| ETP SBS | 47.38 | 9.39 | 10.39 | 2.19 | 11.50 | 3.81 | 14.37 | 0.97 |
| ETP IR | 5.00 | 7.97 | 58.98 | 13.19 | 7.00 | 3.66 | 2.97 | 1.22 |

[a] Aliphatic (Cal), aromatic (Ph), anomeric (OCO), carboxylic and/or amide (COX), ketone (CO) carbon, and carbon bonded to amine (NR), methoxy (OMe), alkoxy (OR), and phenolic and/or phenoxy (PhOY) groups.

**Table 5.** Plants cultivated with SBS and SBS effects: increase (*w/w*% relative to control) of total biomass or crop production, unless otherwise indicated. Data for specific indicators in Section 3.

| | CVDF | CVD | CV | D | ETP |
|---|---|---|---|---|---|
| Euphorbia [15] | 331 | | | 117 | |
| Lantana [16] | 143 | | | 85 | |
| Hibiscus [13,17] [a] | | | 15 | 23 | |
| Murraya [18] | 67 | | | 35 | |
| Tomato Micro-Tom [19] [b] | | | | | |
| *Tomato Lycopersicon* [7,20] [c] | 16 | 4–13 | 21 | 5 | |
| Tomato Micro-Tom [21] | 46 | | 1 | 16 | |
| Red pepper [22] | | 66 | | | |
| Maize [23] | 89 | | | | |
| Bean [24] | | | | | 109–1750 [d,e] |
| Grain [21] | 10 | | 9 | 9 | |
| Tobacco [21] | 6 | | 0 | 0 | |
| Spinach [25] | | | | 24–40 [f] | |
| Oilseed Rape [10] | 56 [g] | | | 42 [g] | |

[a] Reference 13 for CV and 17 for D. [b] Used only as model plant (see below for results). [c] Reference 20 for CVD and 21 for CVDF, CV and D. [d] Increase of enzyme activities and soluble proteins concentration in leaves and roots. [e] Increase of root diameter (66%) by ETP PFB, SBS and IR, and of chlorophyll b (135%) by ETP SBS and IR. [f] Reduction of nitric to total N ratio in leaves. [g] Reduction of plant lesions due to Leptosphaeria maculans.

### 3. Demonstration of SBS Effects as Biostimulant/Photosensitizers in the Cultivation of Tomato, Bean, Euphorbia, Lantana and Hibiscus Plants

*3.1. Tomato Solanum Lycopersicum*

The first trials to investigate SBS performance in the cultivation of food plants were reported in 2012 [7]. The tomato *Solanum Lycopersicum* was used as a probe plant. The plant was cultivated in a farm greenhouse using a commercial product obtained from animal residues (RCP) as organic fertilizer. The experimental plan was designed to measure the plant growth, and the crop production and quality in the cultivation soil treated with RCP (the control treatment), in comparison with the soil treated alternatively with CVD SBS, IR and PFB. Due to the different composition of the control RCP fertilizer and the SBS test materials, the four materials were applied in different amounts to the soil in order to contribute to each soil plot the same 1.1–1.2 organic matter t/ha dose.

The results showed that the plants grown on the CVD SBS treated soil performed better than all others. The former ones exhibited 5–19 day earlier tomato ripening, 4–13% higher production of per plant fruit and per cluster number of fruits, and 7–8% higher leaves chlorophyll content. This result was achieved in spite of the fact that RCP contribute more organic N to the cultivation soil (120 N kg/ha) than CVD SBS (80 N kg/ha). The superior effect of CVD SBS was ascribed to two specific product properties. First, the CVD SBS highest water solubility allowed faster nutrient uptake by the plant, compared to RCP and the other two CVD PCB and IR products. Secondly, CVD SBS had a peculiar photosensitizing properties. These properties had been reported in previous work where the SBS was used to promote the photo oxidation in industrial organic waste waters [26]. In the case of the above tomato cultivation trials, the highest leaf chlorophyll content for the plant cultivated in the CVD SBS indicated enhanced photosynthesis, compared to that taking place in the plants cultivated in the soil treated with RCP, and with CVD PCB and IR. The results of the tomato cultivation trials, coupled to those obtained for the photo oxidation in industrial organic waste water, suggested the fascinating belief that SBS might promote either C fixation or mineralization, according to operating conditions.

Further work was performed by cultivating tomato plants with the D SBS, CV SBS and CVDF SBS [20] described in Tables 1–4. The cultivation trials were carried out in the same farm and conditions as the CVD trials [7], except for the fact that the applied D SBS, CV SBS and CVDF SBS doses were much lower and at three levels: 30 kg/ha, 145 kg/ha and 500 kg/ha [20]. The organic matter doses applied with the D, CV and CVDF SBS ranged from 22–25 kg/ha to 360–420 kg/ha, from 46 to 3 times lower than the organic matter applied with the CVD SBS in the previous work [7]. The control soil in the D, CV, CVDF SBS trials was the same as for the CVD SBS trials. Compared to control, the following most significant effects were measured for the plants grown on the soil treated with SBS: 9.4% increase of plant diameter by 145 kg/ha CV SBS; 9.7% increase of chlorophyll content by 500 kg/ha CV SBS, not significantly different from the increase by the 145 kg/ha treatment; 21% increase of fruit yield by145 kg/ha CV SBS. Generally, for all other SBS treatments and for the control soil, plant performance resulted not significantly different from or lower than that measured for the 145 kg/ha CV SBS treated soil. Compared to the CVD SBS trials carried out at 1.1 organic matter applied dose [7], the 104 kg/ha CV SBS organic matter dose applied in the 145 kg/ha CV SBS treatment [20] gave almost double crop production increase, relative to the control plants. At this regard, the data in Table 6 show that the applied doses of the plant nutrient N, P, K element with the different CVDF, D and CV SBS treatments are far lower than those applied with the CVD SBS [7] treatment, and that the doses applied with the CV SBS are the lowest ones. Yet, the 145 kg/ha CV SBS treatment produced the best performing plants. This fact added further argument to support the belief that the SBS activity was not merely fertilizing the soil with mineral nutrient, but most likely stimulating the plant metabolism. The analysis of the results [7] indicated the CV SBS as the most efficient biostimulant, due to its specific organo-mineral composition and solubility properties.

**Table 6.** Supplied mineral elements (ME) and soil nutrients (kg/ha) in cultivation trials with CVD SBS 1.1 t/ha organic matter [7] and CV, CVDF, D SBS 145 kg/ha dose [20].

|          | Total ME | Total NPK | N    | P    | K    |
|----------|----------|-----------|------|------|------|
| CVDF SBS | 15.5     | 15.2      | 6.29 | 0.91 | 7.96 |
| D SBS    | 17.8     | 25.4      | 11.4 | 0.72 | 13.3 |
| CVD SBS  | 221      | 162       | 80   | 7.4  | 75   |
| CV SBS   | 21.5     | 12.7      | 5.8  | 0.33 | 5.2  |

*3.2. Red Pepper Capsicum Annuum, F1 Barocco*

The CVD SBS was tested also in the cultivation of red pepper [22]. Compared to the first previous tomato study [7], in the cultivation of red pepper [22] the same CVD SBS and soil were used, but CVD SBS was applied to the soil at much lower doses. These were 7, 35, 70, 140, 350, and 700 kg/ha, corresponding to organic matter ranging from 5 to 500 kg/ha. The SBS organic matter application range in the red pepper study [22] was slightly wider than that tested for the CVDF, D and CV SBS in the second tomato study [20], but still from 46 to 3 times lower than that applied with the CVD SBS in the first tomato study [7].

In the red pepper study [22], plant size, leaves' chlorophyll content and crop production over the growing cycle were measured. Compared to the control cultivation, the most significant effect in the presence of CVD SBS were shown at 140 kg/ha applied dose from the leaves' chlorophyll content and fruit production. The leaves chlorophyll content reached a peak upon increasing the SBS dose up to 140 kg/ha and then decreased upon increasing further the SBS dose to 700 kg/ha. Relatively to the control plant, for the plants grown in the 140 kg/ha treated soil the following increases were observed: 12% for the chlorophyll content, 90% for the 1st harvesting week crop, 66% for the total crop production, 17% for the per fruit weight. These effects were far higher than those obtained for the tomato plant cultivations [7,20].

Particularly interesting in the red pepper study [22] was the trend of the leaf chlorophyll content and crop production, both showing a peak at 140 kg/ha CVD SBS applied to the soil. The same trend was observed for the photodegradation of organic pollutants in the presence of different concentrations of SBS [27,28]. In this case, the provided explanation was that the added SBS to the polluted waste water catalysed the photo degradation of the organic pollutant up to a peak added content. At higher SBS concentration, the self-photo degradation of SBS occurred. This lowered the concentration of the pristine SBS, and therefore its availability to catalyse the photo degradation of the organic pollutant in the waste water. In the case of the red pepper study [22], the data confirm the correlation of chlorophyll formation, photosynthesis and crop production. However, understanding the mode of action of the CVD SBS in the biochemical system under investigation is far harder than in the sterile homogenous aqueous system of the industrial waste water [27], where no biochemical reaction took place. By comparison, in the red pepper cultivation system [22], both chemical and biochemical reactions took place. There, the complex processes of photosynthesis and chlorophyll formation was regulated by the availability of enzymes and light. In this respect, the red pepper study [22] did not provide data that could support the effect of the CVD SBS on the natural enzymatic pool present in the system.

*3.3. Bean Phaseolus Vulgaris*

The biochemical response to the SBS was addressed in the study of the effects of ETP SBS (Tables 1–6) on bean plants [24]. Specifically, the plant nitrogen metabolism was studied by determining the nitrate reductase, glutamine synthetase, and glutamate synthase activities and their relation with the content of soluble proteins in the plant leaves and roots. In the bean study, the ETP SBS, ETP PFB and ETP IR were applied separately to a substrate consisting of peat and sand in 14 cm × 14 cm × 15 cm pots. Four grams ETP SBS per pot were applied. Eight grams of ETP PFB and IR per pot were applied.

In this fashion, the three ETP materials contributed nearly equal N and C content to the substrate. The results showed no statistically significant effect on the plant shoot and root weights and leaves' chlorophyll content by the substrate treatments, compared to the control substrate. However, the leaves and roots of the plants grown on the ETP SBS treated substrate exhibited the highest enzyme activities, compared to the control and the other treatments. The increases of enzymes content by ETP SBS were remarkably high. Relatively to the control plant, they ranged from 109 to1750%. The content of the soluble proteins in the plant leaves and roots was also measured. Consistently with the data for the enzyme activities, the plant grown on the substrate treated with ETP SBS had the highest protein content. Relatively to the control plant, the protein increases were 77% in the leaves and 226% in the roots. The data demonstrated that the ETP SBS promoted the highest N assimilation by the plant. This fact was proposed as a possible indication of auxin-like effect by ETP SBS.

### 3.4. Euphorbia x lomi Rauh

Further studies were carried out on the SBS in comparison with commercial products claimed by the vendors as natural organic amendments or biostimulants. The D SBS and CVDF SBS were tested for the cultivation of *Euphorbia x lomi* Rauh [15] in comparison with a Leonardite derived commercial product (LND). The plants cultivation was carried out in pots containing a substrate of sphagnum peat and perlite. Compared to the SBS, the chemical composition of LND was qualitatively similar, but quantitatively very different. The latter had much lower N and P, and much higher K than the SBS. Under these circumstance, the authors chose to apply the LND at the dose recommended by the vendor, and the SBS at the nearly equal weight dose of the as-purchased LND. Coherently with this criterion, the three products were applied in aqueous solution at the following doses (g/pant): 4.6 for CVDF SBS applied as substrate drench, 3.1 and 1.5 for D SBS applied as substrate drench and foliar spray, respectively, and 1.9 and 0.94 for LND applied as substrate drench and foliar spray, respectively. This dose application scheme allowed comparing the CVDF SBS versus LND at close crude product doses, and the two SBS, one with the other, at the same applied N doses.

The following statistically significant effects were measured in the Euphorbia study [15]. The CVDF SBS treatment yielded the highest number of leaves per plant, leaf area, number of flowers per plant, total stem, leaves, roots biomass production, water use efficiency, leaf chlorophyll content and gas exchange than the control plant and the D SBS and LND treatments. Increases relative to the control plants were: 95% for the number of leaves, 78% for the leaf area, 233% for the number of flowers, 331% for total biomass production and water use efficiency, 33% for leaves' chlorophyll content, and 258% for leaf gas exchange. The other treatments also gave significantly higher increases relative to the control plants, but lower increases compared to CVDF SBS. The different effects arose from the different applied products, regardless of the application modes being by foliar spray or substrate application. For these reasons, the highest effects of CVDF SBS could lie on the highest supplied N per plant and the highest content Fe, Ca, P, carboxylic, phenolic and amino groups. Fe ions could have an important role for the plant photosynthesis. By virtue of its organic acid and basic groups, the CVDF kept Fe ions in solution at circumneutral pH. These were inferred responsible for the photosensitizing properties of SBS [21,27,28].

### 3.5. Lantana Camara and L. sellowiana

The results obtained in the Euphorbia study were replicated for the cultivation of Lantana [16] in the presence of the same CVDF and D SBS, and LND products. Two different plant species, *Lantana camara* (CAM) and *L. sellowiana* (SEL), were cultivated. The same ranking order of performance by the tested products was reported for the two plant species. Relative to the control plants, the increases of the measured indicators by the CVDF SBS were: 45% for CAM and SEL plant height; 92% for CAM and 105% for SEL number of leaves; 234% for CAM and 171% for leaf area; 176% for CAM and 326%

for SEL number of flowers; 35% for CAM and 25% for SEL root length; 184% for CAM and 176% for SEL root dry weight; 101% for CAM and 114% for SEL stem and leaves total dry weight; 140% for CAM and SEL water use efficiency; 31% for CAM and 26% for SEL leaf chlorophyll content; 190% for CAM and 181% for SEL leaf gas exchange. The other treatments gave also significantly higher increases relatively to the control plants, but lower increases compared to CVDF SBS. The greater effect of CVDF SBS on the plant photosynthesis and leaf chlorophyll content of the Lantana plants, compared to the plants treated with D SBS and with LND, could be related to the higher content of Si, Mg, Fe and N in the CVDF SBS. Indeed, it is known that Fe and Mg play significant direct roles in photosynthesis, whereas N is present in chlorophyll molecules. By virtue of the organic acid and basic functional groups bonding and/or complexing the mineral elements, the CVDF SBS improved the plant take up and availability of the mineral elements necessary for chlorophyll biosynthesis and photosynthetic activity.

### 3.6. Hibiscus Moscheutos L. Subsp. Hibiscus Palustris

Other authors [13,17] applied the D and CV SBS in the cultivation of Hibiscus. In the first study [17], the D and CV SBS were compared with the D and CV PFB and IR materials, and with a commercial biostimulant (CB). According to the description of the vendor, CB was a plant extract containing fulvic and humic substances, amino acids and glycine betaine claimed to perform as biostimulants. As a consequence of the different products' sources, CB contained 74% organic matter, 24% C and 5.3% N, against 41–66% organic matter, 23–39% C and 1.4–6.6% N for the CV and D materials. Due to the large difference in the chemical composition of the investigated products, different crude product doses were applied in order to guarantee that the amount of added organic carbon contributed by the D and CV products in the cultivation substrate was the possible closest to the amount of C (0.42 kg/m$^3$) contributed by the commercial product as suggested by its vendor. The control (no added SBS or CB) and treated substrates received the same basic standard mineral fertilization. Relevant for this study is the fact that the applied doses of the tested products contributed to the substrate 5–15% of the minimum dose for common organic fertilizers normally applied in agriculture.

The results of the first Hibiscus study [17] showed that all treated substrates gave significant increases of the plant biomass production and biometric parameters, compared to the control substrate. The D SBS treatment gave the highest effects. It yielded 22–33% increases for the dry weights of leaf, stem and flower, total shoot, and leaf, leaf area index. The increase of the leaf chlorophyll was 8% and not statistically significant. The other treatments gave equal or lower increases, compared to the D SBS treatment. The treatments ranking order for the gas exchange activity order was different. In this case, the three CV products gave the highest statistically significant increases, compared to the control substrate: 24% for the photosynthetic activity rate, 46% for stomatal conductance, 31% for the evapotranspiration rate.

Based on the results of the studies on tomato [7,20], Euphorbia [15] and Lantana [16], in the above Hibiscus study [17] two important issues were discussed. First, the Hibiscus data further support the properties of the municipal biowaste derived products as photosensitizers, promoters of photosynthetic activity. Secondly, when products are applied to the cultivation soil or substrate previously treated with the same basic standard mineral fertilization, the mineral composition differences among the added products are likely to be levelled out by the higher relatively amount of nutrients supplied by the conventional chemical fertilizers. Thus, the relationship of the observed effects with the applied products chemical nature and/or composition is likely to be dimmed.

To challenge more the performance of the CV and D SBS as biostimulants, the second Hibiscus study [13] was carried out under nutrient stress conditions. Pot trials were performed a substrate containing peat and pumice at pH 6 by calcium carbonate. Osmocote was used as controlled release fertilizer (CRF). The experimental plan comprised four substrate treatments: the standard fertilization (SF) treatment at 6 kg/m$^3$ CRF dose, and

the low fertilization (LF), the LF with added D SBS (LFD) and the LF with added CV SBS (LFCV) treatments, all last three at 3 kg/m$^3$ CRF dose. The nutrients' supply (kg/m$^3$) was 1.2 N, 0.8 P$_2$O$_5$, 0.7 K$_2$O for SF, and 0.6 N, 0.3 P$_2$O$_5$, 0.3 K$_2$O for LF, LFD and LFCV. Plant performance indicators were biomass accumulation, biometric parameters, leaf gaseous exchanges and elemental composition, and nitrogen (N)-use efficiency.

As expected, relatively to SF treatment, the performance indicators of the plant grown in the LF substrate were measured significantly lower by 47% for plant dry weight, 19% for plant height, 58% for plant volume, 46% for leaf area, 17% for relative growth rate, 22% for N assimilation rate, 23% for the photosynthetic rate, 27% for the chlorophyll content, 60% for the stomatal conductance, and 50% for the evapotranspiration rate. However, the values of the plant indicators measured for the plants cultivated in the LFCV treated substrate were significantly different from those of the plant grown in the SF. The LFD substrate performed significantly better than the LF substrate, but not as well as LFCV. Only in the case of the evapotranspiration rate, the measured values for the plants grown in both the LFD and LFCV treated substrates resulted significantly higher by 35%, relative to the values measured for the SF plants. The treatments significantly also affected the N use efficiency indexes. The LFD treatment gave 17% higher N physiological use efficiency than the other treatments. Compared to the SF and LF treatments, the LFD and LFCV treatments, respectively, enhanced the agronomic N use efficiency by 62% and 117%, and the N recovery use efficiency by 50 and 134%.

The second Hibiscus study [13] established that the Hibiscus plant performance is negatively affected by the low nutrient doses applied in the LF treatment, compared to the SF treatment. It also confirmed that the negative effect of the LF treatment could be well compensated by the SBS addition to the substrate, particularly in the case of the LFCV treatment. Since, the added SBS in the LF substrate did not alter the N, P, and K nutrient content, the positive effects of the added SBS could only arise from their property to stimulate the plant metabolism. This result supported the auxin-like effect in the bean cultivation study [24] proposed for the ETP SBS.

## 4. Replicability of SBS Effects in the Cultivation of Other Food and Ornamental Plants

### 4.1. Murraya Paniculata L. Jacq

*Murraya paniculata* L. Jacq was cultivated [18] under similar experimental conditions as reported for the Euphorbia [15] and Lantana studies [16]. In the Murraya study, the D SBS, CVDF SBS, and LND commercial product were applied at 3.1, 4.5 and 2 g per plant, respectively. The CVDF SBS treatments resulted the most efficient. Relative to the control plant, the plant grown in the CVDF SBS substrate exhibited increases of 61% in plant height, 72% number of stems, 116% number of flowers, 242% number of fruits, 63% number of leaves, 95% leaf area, 67% total stem, leaf, root dry weight, 54% root length, 147% water use efficiency, 88% leaf chlorophyll content, 196% net photosynthesis, and 933% stomatal conductance. The D SBS and LND treatments ranked second and third, respectively, in the order of decreasing efficiency. The products ranking order and the observed increases replicated the results obtained in the previous Euphorbia [15] and Lantana studies [16]. The strong correlation between the plant biometric parameters and the leaf chlorophyll content, net photosynthesis and stomatal conductance, particularly evidenced for the CVDF SBS treatment, strongly supports the hypothesis of the product "auxin-like effect", which was demonstrated in the previous hibiscus [17] and bean [24] studies, respectively, for the D and CV, and for the ETP SBS.

### 4.2. Tomato cv. Microtom, Grain cv. Abate and Tobacco cv. Burley

The CV, CVDF and D SBS were also tested [21] in the cultivation tomato *Micro-Tom*, a model cultivar for plant research [19]. The neat SBS were applied in pots of 15 cm diameter at dose of 240 mg/pot corresponding to 140 kg/ha as the dose tested in the cultivation of tomato *Lycopersicon* in green house farm soil [20]. The experimental plant also included pots containing mixtures of SBS and NPK 20-20-20 mineral commercial fertilizer applied

at 7 NPK/SBS ratio. The control was a sterile substrate. The D SBS + NPK and CV SBS + NPK treatments gave, respectively, 53% and 79% fruit production increment relative to the control soil, followed by 46% increment by the plain CVDF SBS, 40% increment by the plain NPK, 16% by the plain D SBS and CVDF SBS + NPK, and 1% plain CV SBS treatments. The same experiments were performed for the cultivation of grain and tobacco with the plain SBS only. Production increments were much lower: 10% for grain by the three SBS, 6% for tobacco only by CVDF SBS and no increment by the other two treatments.

Comparing the plain SBS treatments in the tomato *Microtom* study [21] to those with the same doses of SBS in the tomato *Lycopersicon* study [20], the former ones gave much higher fruit production. Additionally, in the *Micro-Tom* study [21], the CVDF SBS was the most efficient (46% increment), whereas in the *Lycopersicon* study the CV SBS was the most efficient (21%). These results and those obtained in grain and tobacco cultivation [21] evidence how SBS effects strongly depend on the type of cultivar; i.e., different SBS produce different effects in different cultivars. The high *Micro-Tom* fruit production increments by the D SBS + NPK and CV SBS + NPK treatments reveals a strong synergy between SBS and NPK mineral nutrients. This arises most likely from the biostimulant properties of SBS coupled to their capacity to transfer faster and more efficiently the mineral nutrients in soluble readily available form from the cultivation substrate to the plant. Particularly relevant in the *Micro-Tom* study is the higher performance of plain CVDF SBS compared to plain NPK, although the latter is applied at a dose seven times higher than the CVDF SBS. This is in line with the SBS biostimulant properties demonstrated in the studies on the food and ornamental plants reported in Section 3.

### 4.3. Zea Mays Maize

The CVDF SBS performance was also tested in the cultivation of maize [23], in comparison with urea, CVDF PFB and IR. The study was carried out in a farm in the province of Torino (Italy). The plants were grown in a non-irrigated silty-loamy sol in the summer season. Before seeding, the soil was fertilized with N-P-K (15-15-15) fertilizer at 260 kg/ha dose to each parcel. The three SBS materials were applied to the farm soil at 7–9078 dry matter kg/ha. Urea was applied at 200 kg/ha dose.

The results of the maize trials [23] performed in the farm field showed that all treatments gave significant large increases of kernel production, compared to the control untreated soil. The highest 89% increase were recorded for the CVDF SBS treatment at 50 kg/ha dose. The urea 200 kg/ha treatment gave 38% increase. The other treatments gave increases, which were lower than, although not significantly different from the CVDF SBS 50 kg/ha treatment. Remarkably, the plants grown in the soil treated with 7 kg/ha CVDF SBS dose exhibited the highest photosynthetic activity. Compared to the results obtained in the tomato [7,20] and red pepper [22] cultivation studies, where the highest crop production increases were obtained at 140 kg/ha SBS doses, in the maize study [23] the most efficient SBS dose was lower by almost 3× factor. For all practical purposes, the most remarkable result was the demonstration that 50 kg/ha CVDF SBS yielded higher crop production than 200 kg/ha urea. This prospected high environmental and economic benefits for farmers deriving from the substitution of the commercial urea fertiliser with CVDF SBS. The same perspectives were offered by the tomato *Micro-Tom* study [21], which pointed out the higher performance of CVDF SBS applied at dose seven times lower than the commercial NPK fertilisers.

## 5. Other Effects of SBS in Agriculture: Healthy Plants and Food Crop Production

Two other studies carried out for the cultivation of spinach [25] and oil seed rape [10] have disclosed other important effects of SBS, which are associated to the bio-stimulant properties.

### 5.1. Spinacia Oleracea L. "Gigante d'Inverno"

For the spinach studies, composite materials pellets containing D-SBS, sun flower protein concentrate (SPC) and urea were fabricated and tested [25] for their performances

as controlled release fertilizers (CRFs). The reason was that urea is a largely used fertilizers worldwide. However, it has negative environmental effects due to release of excess nitrogen over the plant uptake rate. In soil urea is hydrolysed to ammonia and then transformed into nitrates, which accumulate in soil and the plant leaves. Excess nitrates leach from soil into ground water and cause eutrophication. Excess nitrates in food plants may have carcinogenic effects for humans. To overcome these drawbacks, CRFs are commercialized. A typical largely used CRF is Osmocote in form of granules containing urea coated with synthetic polymers. The composite D SBS-SPC-urea were fabricated and tested as biobased CRFs materials, which could potentially substitute synthetic organic materials derived from fossil sources.

The first study [25] disclosed the D-SBS property to retard the formation of ammonia from urea hydrolysis and enhance the release of organic nitrogen from SPC. This effect was explained to derive from a plausible chemical interaction of D-SBS functional groups with urea and SPC. As these findings prospected the D-SBS-SPC-urea composites as potential new biobased CRFs, the second study [25] was undertaken to test the above composites in the cultivation of spinach. In this case, a commercial material (Evergreen TS) was used as cultivation substrate. Several formulations containing different amounts of D-SBS, SPC and urea were tested, in comparison with neat D-SBS, SPC, urea, and Osmocote. The cultivation substrate containing no added products was used as control. The trials were carried out in 2 L pots. The test pots contained the same 280–285 mg amount of total N, against 28 mg in the control pot. The plant weight, leaf chlorophyll content, total N and nitrate uptake in leaves and roots were determined at the end of the cultivation trials.

The results of the spinach study [25] showed no statistically significant differences by the substrate treatments compared to the control substrate in leaves and roots weight. The neat SPC and SPC-urea pellets gave significantly the highest leaf chlorophyll content, compared to the control substrate. The other treatments gave lower or not significantly different values, compared to the neat SPC and SPC-urea treatment. The most relevant results were for the nitrate content and nitrate/total N ratio in leaves. The leaves of the plants grown in the substrate treated with the pellets containing D-SBS together SPC and urea had high total N uptake with significantly lower nitric to total N ratio (9.6–12.0), compared to that (15.3–16.5) for the plant grown in the substrates treated with the pellets containing SPC and/or urea, but no D-SBS. The best plants containing high total N content and low nitrates accumulation were those grown in the substrates treated with the SPC-BP, SPC-BP-U, urea-BP and Osmocote® formulation. The nitrate concentration in the spinach leaves of all these plant was below the limit of 2 g/kg recommended for preserved frozen spinach by the European Commission. The results confirmed that all composites containing D-SBS yield the safest crop coupled with high biomass production. These findings proved that, although not supplying the plant as much nitrogen as SPC and urea, D-SBS strongly affects the process of organic nitrogen mineralization in soil. Based on the results of the two studies [25] carried out on the D-SBS-SPC-urea composites, a reaction scheme was proposed encompassing the biochemical and chemical interaction properties of D-SBS.

*5.2. Oilseed Rape Brassica napus L. cv. Columbus*

The oil seed rape study [10] revealed the properties of D and CVDF SBS as plant disease suppressant. Oilseed rape (*Brassica napus* L.) cv. Columbus plants were infected with the fungal pathogen *Leptosphaeria maculans*. Plant cotyledons and roots were sprayed with 2 and 0.02% SBS aqueous solutions, respectively. For comparison with the SBS, Benzothiadiazole (BTH) commercialized by Syngenta under the trade name Bion®, a widely used plant disease suppressant, was also applied at the dose suggested be the vendor. Compared to the control plan (no applied SBS or BTH), the plants treated with SBS showed the following effects. The 2% D and CVDF SBS solutions caused 42 e 56% lower leaf necrosis, respectively. The 0.02% D and CVDF SBS solutions caused 31 and 37% lower leaf necrosis, respectively. By comparison, the BTH treatment caused 80–90% lower leaf necrosis, compared to the control. The study assessed that the SBS induced plant defence by ethylene dependent

signalling pathway. The results showed that the SBS effects were lower than BTH's. On the other hand, the SBS are bio-based products. On this ground, the study pointed out that, in spite their lower effects compared to BTH's, the SBS were environmentally suitable for utilization in organic farming, whereas synthetic chemicals as BTH are not.

The findings of the oilseed rape study [10] spurred further R&D to produce SBS with empowered antifungal properties. The D SBS was oxidized [28] to yield the Dox SBS. Antimicrobial assays are being carried out to assess the Dox SBS power to reduce the mycelial growth of nine targeted fungal phyto-pathogens, which represent serious threats for food and ornamental plants. The efficiency of SBS as potential plant disease suppressant, coupled to their properties as plant growth bio-stimulants and regulators of mineral nutrients release in CRFs, prospect new farming practices with high environmental and economic benefits.

## 6. SBS Economic and Environmental Benefits, and Perspectives for Agriculture and Horticulture

Mineral and organic products are marketed as fertilizers, plant biostimulants, and plant disease suppressing agents. Prices of these products cover a wide range. Benzothiadiazole, used as plant disease suppressants [29], is the most expensive product. Its price is at 800 USD/kg level [30]. By comparison, the production cost of mineral fertilizers is in the 0.11–0.46 €/range. The increasing demand of mineral fertilizers depletes fossil sources. The excessive applied doses to boost crop production causes accumulation in and leaching through soil into natural waters, and consequent eutrophication. In the last few decades, biostimulants have emerged as a new product category for agriculture [31]. This category includes substance or microorganism that, regardless of their mineral nutrients content, are supposed to enhance plant nutrition efficiency, abiotic stress tolerance and/or crop quality traits. They are supposed to modify the plant physiology, and so to enhance the plant growth and stress response. Compared with biofertilizers, biostimulants act at much lower applied doses. Humic substances (HS), extracted from soil and fossil deposits, belong to the biostimulants' category.

The SBS, described in Sections 3–5 of the present review, bear similar origin and chemical features as humic substances [32]. The advantages of SBS compared to HS and other commercial products claimed or reported in the literature as biostimulants is that the SBS are obtained from municipal biowastes available worldwide [33,34]. They do not cause depletion of soil organic matter or fossil deposits, and their production cost is very low. Thus, new eco-friendly and low cost perspectives are opening for novel SBS-based farm practices to replace and/or decrease mineral fertilizers consumption in agriculture.

At the present time, the market turnover of organic fertilizers is small, compared to the mineral fertilizers'. The US total fertilizer market is around 40 billion USD, with only 60 million USD contributed by organic fertilizers. Prices for various organic fertilizers range [3,35–37] range from 140 USD/t for solid products containing 10% soluble organics to 3000 USD/t for products sold in solution containing 35% organics and other mineral elements. Based on information collected by the authors of the present review, through interviews with major Italian distributors of peat derived organic fertilizers, the European market turnover is 20–25 million EUR/year., the minimum sale price is 1000 EUR/t, equivalent to 20–25 kt/year. sale. By comparison, the Euphorbia [15], Lantana [16] and Murraya [18] studies demonstrate that SBS are more efficient biostimulants than commercial products derived from Leonardite. The latter products containing 30% dry matter are sold for 7 EUR/kg [15], which corresponds to over 23 EUR/kg dry matter. The SBS production cost has been estimated about 0.1–0.5 EUR/kg [32]. The figures prospect attracting economic benefits deriving from the allocation of SBS in the organic fertilizer market. Further commercial opportunity for SBS may derive from the growth of the bio-stimulants market [38,39], estimated to reach 5 billion euros in the current decade.

To fully appreciate the economic perspectives of marketing SBS in biostimulants' product category, it should be considered that SBS contain all mineral nutrients needed by

plants (see Section 2 above). These are bonded to the soluble lignocellulosic matter. The research results (see Sections 3–5) point out that the reason of the observed effects on plant growth and productivity is that the SBS supply the plants with the mineral nutrients in a readily available soluble form, thus facilitating the nutrients uptake by the plant. Thus, the SBS fall into the high price organic fertilizers' category. It is also important to be aware of the following fact exemplified for the Italian market. The SBS are obtained from composted urban bio-wastes. Italy produces 4.2 million t/year. organic humid bio-waste [40]. This can potentially yield 300–400 kt/year. SBS. This potential production exceeds the above estimated organic fertilizers market size. It is evident that, at the present time, this market cannot absorb all organic fertilizers that can be obtained from the produced compost.

It should also be considered that the SBS have been proven efficient plant disease suppressants (see Section 5). The capacity to induce plant protection against pathogens adds significant higher value to the potential SBS market [30], in comparison with fertilizers that only enhance plant growth [36,37], but do not have at the same time antifungal properties.

The above literature survey however points out that the organic fertilizers market is in the early stage. In this context, the SBS might be favoured for their capability to provide an integrated complete plant nourishment, which contains both mineral and organic matter of renewable sources. In principle, these products could replace current commercial mineral and organic fertilizers, and also antifungal agents. To appreciate the full potential of SBS uses in agriculture, it should be taken also in consideration the work [41–44] reporting SBS as potential components of new composite mulch films. Used in agriculture, these films might have multiple function, i.e., protecting plants against negative external influences, creating an ideal microclimate, and slowly releasing the SBS into the soil to stimulate plant and crop growth.

Environmental benefits from using of SBS derive mainly from the substitution of mineral fertilizers. The tomato Micro-Tom [21] and maize [23] cultivation studies showed that performance-wise 1 kg SBS is equivalent to 5–7 kg NPK fertilizers. The Euphorbia [15], Lantana [16] and Murraya [18] studies showed that 1 kg SBS yields equal or better plant productivity of at least 1 kg of organic fertilizers derived from fossil source. On this basis, using 1 kg SBS in place of 5–7 kg mineral fertilizers or 1 kg of organic fertilizers from fossil source would allow large reductions of nitrate leaching into natural waters and 100% $CO_2$ emission in air, respectively.

## 7. Results' Summary and Discussion

Five SBS have been tested as organo-mineral fertilizers in the cultivation of thirteen food and ornamental plants (Table 5). The studies have been carried out in comparison with their IR co-products and the pristine sourcing PFB materials (Tables 2–4), with conventional NPK fertilizers, commercial organo-mineral fertilizers claimed by the vendor for their biostimulant properties, and with synthetic controlled release fertilizers and antifungal agents. The collected data demonstrate the performance of SBS as biostimulants (Section 3) and antifungal agents (Section 5.2), and the replicability of their effects over the different tested plants (Section 4).

The results of the trials described in Sections 3–5 evidenced that the performance ranking order of the applied SBS products depended on the cultivated plant species. Overall, ten not commercial SBS and IR research products, five different PFB pristine sourcing materials, and several commercial fertilizers and biostimulants were used for the cultivation of thirteen plants. For each plant, several performance indicators were measured. The ranking order of the applied products also depended on the plant performance indicators. Under these circumstances, it is not possible to summarize the results of the present review results in form of comprehensive figures. Specific data plots are given in each of the references cited above for each trial. The authors feel that the results of the present review are more easily and clearly summarized in form of the following text.

The first trial carried out for the cultivation of tomato [7] demonstrated that CVD SBS performed better than CVD IR, CVD PFB and the commercial RCP product. The second

tomato trial [20] demonstrated that CV SBS performed better than D and CVDF SBS, and CVD SBS in the first trial, even though the applied N, P and K doses with CV SBS were the lowest ones. Thus, the CV SBS seems the best choice for farmers adopting the SBS-based practice for the cultivation of tomato *Lycopersicon* as part of their business activity. On the other hand, potential stakeholders of the SBS-based practice should take in consideration the results of the trials carried out for the cultivation of tomato Microtom and red pepper. In the former case [21], the CVDF SBS performed better than the CV and CVD SBS. In the case of red pepper [22], the results demonstrated that the plants cultivated in presence of CVD SBS performed much better than tomato Lycopersicon cultivated in the presence of CVD SBS [7], and tomato Lycopersicon cultivated in the presence of CV, D and CVDF SBS [20], even though the CVD SBS dose applied in the red pepper study was much lower than that applied in the first tomato study [7], and the same as the CV, D and CVDF SBS doses applied in the second tomato study [20]. The four case studies [7,20–22] definitely prove that the all three CV, CVD and CVDF SBS obtained from composted biowastes represent the best choices depending on the type of food plant to cultivate.

The trials performed for the cultivation of maize, spinach and of the ornamental plants disclosed a somewhat different product hierarchy, particularly in relation to D SBS that never ranked first in the tomato and red pepper studies. The Euphorbia [15], Lantana [16] and Murray [18] studies showed that the CVDF SBS performed better than the D SBS and the commercial LND product. However, the first Hibiscus study [17] showed that D SBS performed better than D IR and PFB, CV SBS, IR and PFB, and the commercial CB product. The second Hibiscus study [10] showed that under nutrient deficiency conditions, both the CV and D SBS treatments compensated the negative effects of the nutrient deficiency on the plant performance indicators well, and that the CV SBS was more effective than the D SBS treatment. The maize cultivation study [23] showed that CVDF SBS performed better than CVDF IR and PFB. The spinach [26] and oilseed rape [10] studies demonstrated D and/or CVDF SBS as regulators of the N release and oxidation and/or antifungal agents. As for tomato and red pepper [7,20–22], the Euphorbia [15], Lantana [16], Murray [18], maize [23] studies confirmed the optimum performance of the CVDF and CV SBS obtained from the composted biowastes, whereas the hibiscus [10,17], spinach [26] and oilseed rape [10] studies disclosed useful important effects by the D SBS.

Table 5 summarizes the SBS ranking order, based on the indicator that for each plant was mostly affected by the applied SBS. It may be observed that, generally, the CVDF SBS produced the highest increase of total biomass and crop production in all plants, which were cultivated in its presence. The biomass and crop production increase ranged from 6% in the case of tobacco to 331% in the case of Euphorbia. Generally, the ornamental plants were more sensitive to the SBS application than the food plants. The other CV, CVD and D SBS were also effective, although at lower level than the CVDF SBS. The former ones produced biomass and crop production increases, which ranged from zero in the case of tobacco to 117% in the case of Euphorbia cultivated in the presence of D SBS.

Although not as effective as CVDF SBS on biomass and crop production, the D SBS exhibited other relevant effects, such as reduction of nitric to total N ratio in spinach leaves and of lesions caused by Leptosphaeria maculans in oilseed rape. In the spinach trial [25], the D SBS was used as a component of a composite pellet also containing urea and sunflower protein concentrate. Although the SBS was a minor component in the pellet and was not a significant source of N, relatively to N supplied by the other two components, the D SBS was found to cause a number of effects. It slowed down the formation of ammonia from urea hydrolysis and enhanced the release of organic nitrogen from the sunflower protein concentrate in the cultivation substrate, and strongly affected the mineralization of the total N supplied by the pellet. These effects were ascribed to the interaction of the D SBS functional groups with urea and the protein concentrate. The hypothesis is quite plausible considering the relative high content of carboxylate functional group coupled to the other NR, OR and PhOY in the D SBS (Table 4). From the practical point of view,

the final effect of D SBS was 24–40% reduction of nitric to total N ratio in leaves and the production of the safest crop coupled with high biomass production.

The oilseed rape study [10] investigated the mechanism of the disease suppression effect reported by both CVDF and D SBS. The two SBS did not show any antimicrobial effect against *L. maculans* in vitro. This suggested different mechanisms underlying the observed reduction of lesions caused by *Leptosphaeria maculans* reported in Table 5, among which the induced resistance characterized by an increased resistance to infection occurring after a previous pathogen attack.

Quantitatively, the most remarkable effects were exhibited by the ETP SBS. This product produced 109–1750% increase of the of the enzyme activities and soluble proteins concentration in the leaves and roots of bean plants. This effect demonstrated undoubtedly the biostimulant properties of the applied product.

Reasons for the SBS performances have been proposed. These are based on chemical and biochemical interactions/reactions catalysed by SBS thanks to their chemical composition. The SBS contain 15–30% minerals together with organic matter. They can, therefore, add soluble plant nutrients to soil. They can also perform as bio-effectors, stimulate the uptake from roots of soil nutrients with a hormone-like effect and/or plant growth by promoting rhizobacteria, and catalyse the plant photosynthetic activity (see all references in Section 3). Applied in mixtures [22,26] with urea, other organic fertilizers and NPK conventional mineral fertilizers, they can regulate the release of N, P and K to the soil and taken up by the plant, and so reduce the amount of nitrates leaching through the soil into natural waters and taken up by the plant crop.

Whereas from the basic science point of view, the collected data do not allow demonstrating the action mechanism for the observed performances of SBS in agriculture, from the practical point of view, the most relevant result is that the highest SBS effect on the plant performance indicators occurs at about 140 kg/ha applied dose to the cultivation soil or substrate. Depending on the plant and the type of applied SBS, increases up to three order of magnitude for the plant performance indicators are measured relatively to the control plants. At higher dose levels, no further increases are observed. The remarkable high effects occurring at relatively low treatment dose prospect using the SBS to augment plant growth and productivity, and at the same time reduce the consumption and negative environmental impact of conventional fertilizers applied at high dose.

The collected data offer ground to attempt establishing case-by-case correlations between the composition and the effects of the different SBS used for the cultivation of the same plants. However, the SBS are complex mixtures of organic molecules differing for molecular weight and chemical features. These molecules are in turn bonded to different mineral elements. Under these circumstances, correlations between SBS chemical composition and effects as given in Sections 2–5 do not help much to identify the active molecules, which are responsible for the observed effects. These difficulties are inherent to products obtained from materials of biological origin.

## 8. Conclusions

Undoubtedly, the present review offers ground for scientists to pose many questions needing explanations, particularly in relation to the action mechanism underlying the SBS effects. Based on the different compositions of the 15 materials used to cultivated the 13 plant species (Tables 2–5), any scientist is aware that the answer to the many questions required identification and isolation of the active principles in each of the 15 materials, and testing them individually for each plant species. Such task would involve experimental work requiring cost and time, which make rather doubtful achieving success. Under these circumstances, the authors think that further tentative explanations of the many questions posed by the present review would be only speculative. Yet, the data obtained from the authors over the past 11 years are very useful to guide farmers in their new SBS-based farm practices. Whereas the present review leaves many scientific questions answered, it demonstrates the replicability of the SBS effects under different environmental conditions.

In absence of the present review, potential stakeholders should search, read, study and evaluate each of the single papers cited in the reference section of the present review, and try to draw the conclusions that allowed them adopting successfully the SBS-based farm practice. On the other hand, in spite of its mainly empirical nature, the present review offers interesting and useful scope for scientific research.

At the present time, the implementation of SBS to operation level in real farm practices must rely on the empirical findings reviewed in the present paper. To this end, the most important factor is the replicability of the composition and performance of each SBS dedicated to the cultivation of specific plant species. Potential stakeholders may rely on the fact that a waste treatment plant may produce a wide variety of SBS tailored for the cultivation of specific plants, depending on the variety of bio-waste sources and treatments types. Further studies should assess how the variability of the bio-waste source, as a function of the environmental and operational conditions where it is produced, may affect the composition and performance of the intended SBS.

**Author Contributions:** Conceptualization, E.M; data curation, E.M., G.F. and A.B.; writing-original draft preparation, E.M.; writing-review and editing, E.M., G.F and A.B.; funding acquisition, E.M. All authors have read and agreed to the published version of the manuscript.

**Funding:** This review was carried out to establish the baseline for the LIFEEBP LIFE19 ENV/IT/000004 project funded by the European Commission under the LIFE 2019 programme.

**Institutional Review Board Statement:** Not applicable.

**Informed Consent Statement:** Not applicable.

**Data Availability Statement:** Not applicable.

**Conflicts of Interest:** The authors declare no conflict of interest.

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
