# Peer review of "Biostimulant Effects of Waste Derived Biobased Products in the Cultivation of Ornamental and Food Plants"

_agriculture, doi:10.3390/agriculture12070994_

Round 1

Reviewer 1 Report

The Manuscript “Biostimulant Effects of Waste Derived Biobased Products in the Cultivation of Ornamental and Food Plants” requires revision. The specific comments are given below.

1.     Remove the abbreviations from the abstract.

2.     In the abstract, provide the most important numerical results of your research.

3.     In the last paragraph of the introduction, clearly indicate the research hypothesis.

4.     Present the results obtained by other authors in a tabular form.

5.     You are only citing 45 items. Do a broader literature review.

6.     Little cited literature from 2018-2022. Refresh references.

7.     Extend your conclusions with the most important results (numerical values).

Author Response

Line numbers in the authors’ reply refer to the revised manuscript showing track changes.

Authors reply to reviewer 1 points 1 through 7.

  1. Reviewer: abbreviation in abstract. Authors: the SBS abbreviation is too important. It identifies the product used in the agriculture trials throughout the whole paper. It is used five times in the abstract. Its removal would make the abstract longer and more difficult to read.
  2. Reviewer: provide most important numerical results in abstract. Authors: see lines 22-24 in the revised manuscript
  3. Reviewer: last paragraph of introduction. Authors: see lines 89-94 in the revised manuscript
  4. Reviewer: results by other authors in tabular form. Authors: in literature, there are no authors testing the SBS, except those cited in the references.
  5. Reviewer: broader literature review. Authors: in literature, there are no products having the specific solubility properties and chemical features of SBS. The cited references contain hundreds of items derived from exhaustive literature reviews of results obtained with different biostimulants on each plant species reported in the present authors’ review.
  6. Reviewer: little cited literature from 2018 to 2022. Authors: see authors’ reply for points 4 and 5.
  7. Reviewer: extend conclusions. Authors: see new sections 7 and 8 at lines 659-832.c

Reviewer 2 Report

The manuscript Agriculture-1795302 is a review of the works of the authors. The manuscript contains innovative information and very interesting results. However, the document does not seem a scientific manuscript. The text is well written, but the core of the manuscript “only” summarizes the results of the previous manuscripts. There is not a critical review of their work and there are many questions that the authors have to explain. For example, What happens in other similar works? Why one non-commercial product is better than another? Why works?

The current version of this manuscript seems a report of the Life project (whit all my respect). I mean that the manuscript demands more discussion of results and comparison with references.

Other specific comments:

Lines 81 – 82: it´s only an idea, but I think that biostimulants reduce/minimize “the negative environmental impact of the excessive application of commercial fertilisers and pesticides.” Your original sentence involves a “dangerous” idea about the use of fertilizers. When I read your sentence, I thought that the use of fertilizers produces negative effects at any dose. All of us Know that the introduction of nutrients to agricultural soil is mandatory. Therefore, I think that the details in your asseveration are important

Section 2 SB composition and properties: explain in detail the procedure to analyze the samples and the results obtained., moreover table 4

Lines 512 – 515 some of the sentences were used in the introduction

Reviewer 3 Report

line 84 needs reference. Which work exactly the authors are referring to?

the structure of introduction is good, but english can be polished.

the titles of the sub sections of section 3 must be elaborated even more. just "tomato" or just "bean" is not self explanatory. each subtitles have to be understandable on their own.

the above comment applies to section 4 and 5 as well. 

section 7 can be separated into two sections. one containing results and discussions, and the other one containing conclusion. The results and discussion can make use of some figures that explains the findings in more efficient way than just text. 

Round 2

Reviewer 1 Report

Thank you for the answers.

Reviewer 2 Report

The manuscript contains the same gaps as the first version. There is not a critical review, the questions about the mechanisms of action and comparison with commercial products or other publications is really scarce.

Reviewer 3 Report

Comments have been addressed